# Point Mutations of Nicotinic Receptor α1 Subunit Reveal New Molecular Features of G153S Slow-Channel Myasthenia

**DOI:** 10.3390/molecules26051278

**Published:** 2021-02-26

**Authors:** Denis Kudryavtsev, Anastasia Isaeva, Daria Barkova, Ekaterina Spirova, Renata Mukhutdinova, Igor Kasheverov, Victor Tsetlin

**Affiliations:** 1Shemyakin-Ovchinnikov Institute of Bioorganic Chemistry, Russian Academy of Sciences, Miklukho-Maklaya 16/10, 117997 Moscow, Russia; nastya-18.02@yandex.ru (A.I.); katya_spirova@mail.ru (E.S.); renata.mukhutdinova@gmail.com (R.M.); iekash@ibch.ru (I.K.); vits@ibch.ru (V.T.); 2Moscow Institute of Physics and Technology, 141700 Dolgoprudny, Russia; 3Biological Department, Lomonosov Moscow State University, 119991 Moscow, Russia; daria.barkhova@gmail.com; 4Institute of Molecular Medicine, Sechenov First Moscow State Medical University, Trubetskaya str. 8, bld. 2, 119991 Moscow, Russia; 5Institute of Engineering Physics for Biomedicine, MePhi, 115409 Moscow, Russia

**Keywords:** muscle nicotinic receptor, gain-of-function, channelopathy, slow-channel congenital myasthenia, patch-clamp, molecular dynamics, calcium imaging

## Abstract

Slow-channel congenital myasthenic syndromes (SCCMSs) are rare genetic diseases caused by mutations in muscle nicotinic acetylcholine receptor (nAChR) subunits. Most of the known SCCMS-associated mutations localize at the transmembrane region near the ion pore. Only two SCCMS point mutations are at the extracellular domains near the acetylcholine binding site, α1(G153S) being one of them. In this work, a combination of molecular dynamics, targeted mutagenesis, fluorescent Ca^2+^ imaging and patch-clamp electrophysiology has been applied to G153S mutant muscle nAChR to investigate the role of hydrogen bonds formed by Ser 153 with C-loop residues near the acetylcholine-binding site. Introduction of L199T mutation to the C-loop in the vicinity of Ser 153 changed hydrogen bonds distribution, decreased acetylcholine potency (EC_50_ 2607 vs. 146 nM) of the double mutant and decay kinetics of acetylcholine-evoked cytoplasmic Ca^2+^ rise (τ 14.2 ± 0.3 vs. 34.0 ± 0.4 s). These results shed light on molecular mechanisms of nAChR activation-desensitization and on the involvement of such mechanisms in channelopathy genesis.

## 1. Introduction

Muscle nicotinic acetylcholine receptor (nAChR) plays the key role in signal transduction between the nervous system and skeletal muscles [1]. Abnormal functioning of muscle nAChR leads to various pathological conditions. The later include certain cases of myasthenia gravis caused by autoimmune antibodies to muscle nAChR [2] and congenital myasthenic conditions [3,4]. Various mutations of muscle nAChR contribute to about half of the congenital myasthenic conditions [3]. Such conditions are not human-specific but also affect other mammals such as dogs, cats [5], and rodents [1]. Among congenital myasthenic conditions, the group caused by dominant gain-of-function mutations, called slow-channel myasthenic syndromes (SCCMSs), attracted previously our attention [6,7]. SCCMS have long history in translational medicine research. The term “slow-channel” was proposed back in 1988 by Engel basing on the observation of prolonged decay of quantal endplate currents [8]. Since then, several substances with muscle nAChR open-channel blocking activities have been repurposed to fight SCCMS: ephedrine and its analogue salbutamol [2,9,10]; quinidine, quinine, and chloroquine [11,12] and fluoxetine [13]. Despite effectiveness of these therapeutics against SCCMS, all the compounds used are not selective toward muscle nAChR and have their own primary molecular targets (ionic channels and neuromediator transporters) and specific SCCMS drugs remain to be developed.

Interestingly, not all slow-channel mutations are located near the ion pore region. One such mutant, bearing G153S substitution in the α1 nAChR subunit is known for enhanced affinity toward acetylcholine [14]. This mutant, contrary to some other “slow-channels”, does not show higher mean channel open times but is characterized by the decreased acetylcholine dissociation and prolonged open channel bursts [14]. However, patients affected by G153S SCCMS are successfully treated with fluoxetin, which blocks open channel of muscle nAChR [15].

The region in which G153S substitution takes place is called loop B. The glycine residue in this position of loop B is conserved in the nAChR α1, α7, α9, and α10 subunits, the subunits respective receptor subtypes showing low affinity toward acetylcholine, but it is substituted to other residues in nAChRs showing high affinity toward acetylcholine [16]. Gain-of-function mutations (e.g., promoting higher acetylcholine affinity) provoke excessive Ca^2+^ accumulation in the end-plate [17] followed by synaptic degeneration, which finally causes muscular weakness in SCCMS patients [18] and in model animals [19].

Amino acid residue 153 is situated outside the acetylcholine-binding pocket but near the loop C outer surface (see below). Thus, some pathway should exist to conduct the allosteric interaction from serine 153 to agonist-binding pocket and ion channel to explain the action of the mutation situated at a distance. One of the most crucial regions of α subunits is the loop C: the X-ray structures of the acetylcholine-binding protein (AChBP) complexes with nicotine (agonist) and such competitive antagonists as α-cobratoxin, α-bungarotoxin, and α-conotoxin demonstrated that an agonist is embraced by this loop moving to the centre of the molecule, while with antagonists it is shifted to the periphery of molecule [20]. By itself, the loop C and peptide mimics thereof are capable of binding certain nAChR ligands [21,22,23]. Thus, it is feasible to hypothesize that some loop C residues are among the first intermediaries between the G153S mutation and altered ion channel kinetics. In this article we provided the support to the hypothesis that L199 residue located at the loop C outer surface interacts with loop B and, thus conducts the conformational disturbance evoked by G153S mutation to the acetylcholine-binding site.

## 2. Results

### 2.1. Molecular Dynamics of Extracellular Domain in α1(G153S) Mutant

To study the hydrogen bond formation between the residues in the 153 and 199 positions (described previously by [16] for a neuronal heteromeric nAChR) a 5 ns molecular dynamics of α1 nAChR extracellular domain was performed using recently published *Torpedo* (systematic name *Tetronarce californica*) cryo-EM structure [20]. This muscle-type receptor is closely related to human and murine muscle nAChRs (80% protein sequence identity for α1 subunit) and has the same residues in the 153 position and in the C-loop.

Extracellular domain of *Torpedo* α1 nAChR (residues from 1 to 209, see Figure 1a) has been copied to a separate file, energy minimized, equilibrated, and subjected to molecular dynamics procedure as described in the Methods. Mutation G153S has been introduced via the “Rotamers” function of UCSF Chimera and mutated molecule has been subjected to the same molecular dynamics procedure as the wild type (WT) one. All heavy atom trajectories and resulted structures were centered at the protein molecule center of mass and periodic boundary conditions were removed. Structures were visually inspected and analyzed using standard Groningen Machine for Chemical Simulation (GROMACS) instruments to calculate gyration radius, backbone root mean square deviation (RMSD), root mean square fluctuation of C_α_ of amino acid residues (RMSF), and H-bond parameters. RMSF were recorded to the b-factor column of averaged PDB file and molecular surface coloring have been applied to both structures (WT and G153S) revealing differences between the flexibility of the C-loop and the N-terminal regions (Figure 1a).

Both structures showed reasonable stability during the course of molecular dynamics with RMS deviations of the backbone not exceeding 0.2 nm (Figure 1b). However, more close observation of residues RMSF revealed a difference between the WT and G153S flexibility of the N-terminal (10–25) and C-loop (180–200) regions (Figure 1c).

To find out what structural differences could arise upon glycine 153 substitution to serine, H-bonds formed by these residues have been searched along the molecular trajectories. Indeed, it was found that, in contrast to wild-type nAChR in which backbone oxygen of G153 forms periodically H-bond with backbone nitrogen of L199 (Figure 1d), the mutant receptor shows the H-bond formation between S153 side chain oxygen and L199 backbone nitrogen (Figure 1e).

Thus, G153S mutation associated with slow-channel myasthenia modifies hydrogen bonding of the B-loop and C-loop and influences specific amino acid residues flexibility. We hypothesized that a second mutation of L199 to a residue, capable of forming extra hydrogen bonds via the side chain could change the distribution of such bonds. The alternation in H-bonds distribution could in principle influence a functional property of the receptor allowing manipulation of nAChR activity by introducing mutations outside the orthosteric ligand-binding pocket, at the same time conserving the binding pocket residues crucial for the receptor functioning.

### 2.2. Design of Double Mutants to Modify H-Bond Arrangement of 153-rd Residue

Serine 153 side-chain hydroxy group in the SCCMS mutant receptor forms a hydrogen bond with the L199 backbone nitrogen. To modify this pattern a threonine substitution at the 199 position was suggested. Threonine has a hydroxy group capable of H-bond formation and its bulky side-chain with intermediate hydrophobic properties is making it an ideal candidate to substitute leucine (which also has a bulky branched side chain) [24].

To study how threonine introduction to the C-loop of α1 nAChR might influence the hydrogen bonds distribution between residues 153 and 199, point mutants were constructed as described in the previous section. Complementary to L199T mutation, L199A has also been studied to probe possible point mutation effects not related to H-bonding. Molecular dynamics revealed a striking difference among L199T, L199A, G153S/L199A, G153S/L199T, and WT receptors (Figure 2a).

Threonine forms an intraresidue hydrogen bond with oxygen of the peptide bond. The ability to form such H-bonds is a well-known property of threonine residue, contributing to 30% of the side chain to main chain hydrogen bonds formed by this residue [25]. In single L199T mutant (**1**) such an intraresidue hydrogen bond was the only option observed during the course of 5 ns molecular dynamics. No G153 to C-loop “backbone to backbone” H-bonds have been detected in contrast to the WT receptor (see Figure 1d). Leucine 199 mutation to alanine (**2**) has also resulted in structures lacking any H-bonds between G153 and C-loop backbone (Figure 2a) confirming a crucial role of L199 in fine-tuning the receptor conformations.

Combining L199A and G153S mutations (**3**) leads to the formation of the H-bond between S153 side-chain and C-loop backbone, like in the case of a single G153S mutant (shown in Figure 1e). Most noticeably, a combination of L199T and G153S mutations leads to a completely new H-bonds pattern as compared to the WT receptor and single G153S mutant. An introduction of T199 to G153S SCCMS-associated mutant stimulates a formation of H-bonds between S153 and Y198 side-chain hydroxy group (structures **4** and **5** on Figure 2a). C-loop tyrosines of nAChR α subunits form a so-called “aromatic box” in the othosteric binding sites. They are crucial for acetylcholine binding and receptor activation [20]. Surprisingly, no hydrogen bonding between T199 and S153 sidechaines was observed during 5 ns molecular dynamics. This observation could be explained by the influence of intraresidue H-bond formation leading to relatively stable 6-member ring formation by T199 itself (**6** on Figure 2a).

A close examination of hydrogen bonds properties distribution confirms the change upon L199T mutation in SCCMS-associated G153S mutant of nAChR. A comparison of two double mutants—L199A/G153S and L199T/G153S—shows that the donor–acceptor angle distribution in the L199T/G153S mutant was skewed toward higher values whereas donor–acceptor distances were practically the same (Figure 2b). These results mean that second mutation introduced to SCCMS-associated G153S muscle nAChR highlights new receptor properties and provokes substantial conformational changes.

To test the second mutation influence on the receptor function, a set of point mutations was introduced to a murine α1 nAChR subunit expressed in transiently transfected neuro 2a cells with the β1, δ, and ε nAChR subunits.

### 2.3. Fluorescent Ca^2+^ Assay of Mutated Muscle nAChRs

Point mutations to mouse muscle nAChR were introduced as described previously [6]. Calcium fluorescence has been measured using the Case 12 genetically encoded sensor as described by Shelukhina [7] in 96-well plates to evaluate dose–response relationship and using the epifluorescent microscope set-up to study calcium rise amplitudes and exponential decay kinetics (Figure 3).

Acetylcholine has higher potency on SCCMS-associated mutant α1(G153S) than on the WT muscle receptor. In the 96-well plate fluorimeter assay we detected a parallel shift of the dose–response curve and in EC_50_ values from 2503 on WT to 146 nM on G153S (Figure 4a). Introduction of the second mutation to L199 position changes EC_50_ of a double mutant depending on the nature of a substituting residue. Double mutant L199T/G153S has intermediate EC_50_ as compared to WT and solo G153S, whereas solo L199T has EC_50_ almost identical to that of WT. Introduction of solo L199A or double L199A/G153S mutations to the α1 nAChR subunit leads to a change in the dose–response curve slope (Figure 4b), which could be attributed to the change in cooperativity of acetylcholine binding. Acetylcholine potency on all tested receptor variants are shown in Table 1.

It is worthy to note that in the 96-well plate Ca^2+^ assay the receptors containing G153S mutation tend to show submaximal responses to high acetylcholine concentrations (see the rightmost points on Figure 4a,b). Such a type of apparent self-inhibition is well-known for nAChRs (e.g., [26,27]). However, the 96-well plate assay measures the integral signal from the entire well. Single-cell Ca^2+^ flux remains uncharacterized in such experiments. Complementary to dose–response curves obtained in a 96-well plate assay, the amplitudes of fluorescent signals from individual cells were studied using optical microscopy.

Three different acetylcholine concentrations (10, 100, and 200 μM) were applied to neuro 2a cells expressing WT, G153S, L199T, L199A, L199T/G153S, and L199A/G153S. Amplitudes recorded from ten individual cells were averaged and normalized to maximal responses for each mutant (Figure 4c). All nAChR variants except L199A and L199A/G153S showed near-maximal response amplitudes at maximal acetylcholine concentrations in contrast to plate-based fluorescent assay results. Thus decreased responses to high acetylcholine concentrations detected in 96-well plate calcium assay arise not from a decrease of the response amplitude. High acetylcholine concentrations should modify shapes of the responses and net Ca^2+^ flux. L199A and L199A/G153S mutants did not show self inhibition at high acetylcholine concentration at integral 96-well plate responses, but there were signs of response amplitude reduction, in contrast to the L199T and L199T/G153S mutants.

To investigate the influence of high acetylcholine concentration on the Ca^2+^ response shape, fluorescent signal kinetics was measured (Figure 4d). The time constants of the exponential decay of acetylcholine-evoked fluorescent signals recorded from L199T and L199A/G153S were not significantly different from WT signals (Table 1). The G153S mutant showed a significantly prolonged Ca^2+^ rise comparing to any other muscle nAChR variant tested in this work. Note, that the L199A mutant showed slightly prolonged Ca^2+^ rise half-life, comparing to WT and L199A/G153S. However, the L199T/G153S mutant showed unexpectedly faster signal decay than WT, despite the G153S mutation presence.

These results confirm that incorporation of a hydrogen bond-forming residue in position 199 (which by itself does not change any of the properties observed in Ca^2+^ fluorescent assay) to a SCCMS-associated G153S mutant influences the activation and desensitization in the receptor.

A more prolonged cytoplasmic calcium influx of the G153S mutant comparing to WT muscle nAChR leading to Ca^2+^ overload could be the reason of muscle pathology [13,28]. The difference between the SCCMS-associated mutant and wild-type muscle receptor could arise from a modification of agonist binding properties, change in allosteric signal propagation from the agonist-binding site to the ion pore, channel activation/inactivation, and desensitization of the receptor.

### 2.4. Patch-Clamp Investigation of WT and G153S Muscle nAChRs Functional Properties

To investigate ion-conducting properties of WT and mutant muscle nAChR electrophysiology experiments were done on the same transiently transfected neuro 2a cells that were used in the fluorescent Ca^2+^ assay. Currents were recorded during a five-second application of 20 μM acetylcholine, digitized, and exported to ASCII files (Figure 5a).

Amplitudes of acetylcholine-evoked currents were significantly (*p* < 0.05, two-sample *t*-test) higher in G153S than in WT muscle nAChR-transfected cells, although amplitudes of WT receptor mediated currents varied greatly (Figure 5b). Higher amplitudes lead to greater calcium ions accumulation in cytoplasm, which partially explains the significantly slower decay of the calcium signal in the fluorescent assay (Table 1).

Apart from the increase in amplitude, the gain-of-function mutation could have altered desensitization leading to inadequate net charge flow through the receptor ion pore. The analysis was focused only on decay phases because they are dependent on the receptor inherent properties but not on the buffer application system speed, at least in a time scale of seconds. To find out if the exponential decay of acetylcholine-evoked current is altered significantly as a result of G153S mutation, traces were scaled, normalized, and averaged (*n* = 8 in each group). The two-exponential curve fitting revealed unexpectedly a higher desensitization speed of G153S mutant in contrast to WT muscle nAChR (Figure 5c). Thus, G153S SCCMS-associated nAChR mutant shows faster acetylcholine-evoked current decay, which counterbalances higher affinity toward acetylcholine and higher average amplitude. However, when combined, these properties result in prolonged Ca^2+^ transients in G153S-expressing cells.

## 3. Discussion

In this article we explore hydrogen bond interactions between the parts of muscle nicotinic acetylcholine receptor (nAChR) associated with ligand binding and with slow-channel myasthenic syndrome (SCCMS). In particular, we focused on α1(G153S) point mutation, which is known to cause SCCMS in human subjects. Patients with SCCMS develop a fatigue as a result of gain-of-function mutation, leading to higher affinity toward acetylcholine and overaccumulation of calcium ions in sarcoplasm in an uncontrolled manner, which provokes excitotoxic muscle fiber death and a compensatory muscle nAChR run-down [18,29]. It was proposed in the literature that introduction of serine, capable of forming additional hydrogen bonds, in place of glycine 153 lacking any functional groups in the side chain, could modify nAChR conformations and thus ligand-binding and activation properties [14,30].

Using recently published *Torpedo* muscle-type nAChR in complex with α-bungarotoxin cryo-EM structure [20], we performed molecular dynamics computations on the α1 nAChR extracellular domain to observe hydrogen bonds formation between G153 residue and C-loop backbone, which were reported in the literature [16]. Indeed, in such a molecular model based on the cryo-EM structure we observed a backbone-to-backbone interaction through a hydrogen bond, which was disrupted by substitution G153 to serine (Figure 1). Point mutation α1(G153S) is one of only two known gain-of-function mutations, located in the α1 extracellular domain and associated with congenital myasthenia development in humans. G153S mutation is situated in B-loop outside the ligand-binding pocket, but changes significantly the affinity for acetylcholine [30]. 

To better understand the mechanisms underlying the first links in pathology development, we introduced a second mutation into α1(G153S) to counteract the Ser 153 hydrogen bonds with the C-loop. Based on molecular dynamics of in silico generated α1(L199A), α1(L199T), α1(G153S, L199T), and α1(G153S, L199A) mutant molecular models we proposed a putative molecular mechanism of the first-level interactions between the residue involved in SCCMS with C-loop that presumably affect normal nAChR functioning. Molecular dynamics showed that introduction of either alanine or threonine to position 199 (part of C-loop in the vicinity of B-loop and particularly of G153 residue) induces a change in hydrogen bonds distribution, which might lead to subsequent allosteric rearrangements of the receptor (Figure 2), confirming importance of this region to the receptor functioning. Molecular dynamics showed that the G153 residue backbone oxygen in the WT muscle nAChR forms a hydrogen bond with L199 (Figure 1d). Local effects of point mutations in the 199 position include emergence of intraresidue threonine hydrogen bonds (see **1** and **6** on Figure 2a), disruption of pre-existed inter-residue hydrogen bonds (in case of L199A solo mutation; see **2** on Figure 2a) and formation of new inter-residue hydrogen bonds, which were not observed in WT and solo L199A mutant (in case of double L199A/G153S and L199T/G153S mutants; see **3**, **4**, and **5** on Figure 2a). Thus, hydrogen bonding between residues situated in 153 and 199 positions of the muscle nAChR α1 subunit has a complex nature, which exerts diverse effects on the nAChR functioning upon mutation of these residues. We hypothesized that double mutants should demonstrate significantly different acetylcholine potency and channel kinetics and tested this hypothesis in the fluorescent assay and patch-clamp electrophysiology.

Indeed, the Ca^2+^ fluorescent assay performed on single and double mutants confirmed the cross-talk between the 199 and 153 positions. A single L199T mutation did not influence acetylcholine potency in the functional fluorescent assay when placed in the WT muscle nAChR, it changed acetylcholine potency if G153S mutation was present nearby (Figure 4a). EC_50_ value of the L199T/G153S double mutant was four times increased comparing to single G153S mutant (Table 1). L199A mutation either solely or in pair with G153S showed a steeper dose–response curve shape (Figure 4b) and somewhat intermediate EC_50_ values (Table 1). It may be speculated that less bulky alanine residue introduced in the C-loop outer surface modifies conformations of the receptor, but this suggestion needs further investigation. Threonine residue, on the other hand, occupies approximately the same volume as leucine and did not influence the receptor conformations the same way as small alanine (compare EC_50_ values for L199T, L199A, and WT in Table 1). These results combined together speak in favor of our hypothesis about the ability of additional hydrogen bonds to serve as carriers of slight conformational changes influencing agonist binding or allosteric signal propagation from orthosteric ligand-binding site to the ion pore of the receptor.

Another set of the mutated muscle nAChR properties tested in fluorimetric Ca^2+^ assay is agonist autoinhibition at higher concentrations (Figure 4c) and cytoplasmic Ca^2+^ acetylcholine-evoked response decay (Figure 4d). It was reported that higher concentrations of acetylcholine are capable of inhibiting of the receptor, causing decreased responses on the right side of the dose–response curve [26,27]. In our 96-well plate setup we also observed decreased responses to 90 μM acetylcholine in muscle nAChRs bearing G153S mutation (Figure 4a,b). To investigate the Ca^2+^ rise with better temporal resolution than can be achieved in the plate reader we performed measurements of cytoplasmic Ca^2+^ rise amplitude in response to 10, 100, and 200 μM acetylcholine using epifluorescent microscope (Figure 4c). Contrary to the 96-well plate data, we did not observe drastically diminished amplitudes of acetylcholine-evoked responses, although L199A and L199A/G153S mutants tended to show decreased responses to 100 and 200 μM of acetylcholine, respectively (Figure 4c). Thus, we concluded that in our 96-well plate setup apparent acetylcholine autoinhibition in higher micromolar range is not attributed to the fluorescent response amplitude reduction. One possible explanation of apparent acetylcholine autoinhibition is the difference in Ca^2+^ rise kinetics, which, in combination with plate reader technical limitations of fluorescent detection speed, could cause underestimation of the response.

We analyzed averaged recorded Ca^2+^ fluorescent signals to find out if generated point mutants differ from each other and WT muscle nAChR in an ability to provoke cytoplasmic Ca^2+^ elevation. Overaccumulation of calcium ions in cytoplasm is considered to be the reason of excitotoxic effects of SCCMS-associated mutations of muscle nAChR [8,18,31]. In our neuro 2a cell-based model of SCCMS we detected not only higher potency of acetylcholine [7] but also prolonged calcium fluorescent signal decay as compared to WT muscle nAChR (Figure 4d). SCCMS mutant G153S showed the most prolonged calcium fluorescent signal among other tested mutant muscle nAChR (more than a twice slower time constant than that of WT—see Table 1). Double mutant L199T/G153S, which showed the most drastic changes in molecular dynamics experiments and in the dose–response assay, demonstrated the fastest fluorescent Ca^2+^ signal decay among the tested, while other mutants showed intermediate results (Table 1). Thus, fluorescent assay of the calcium transient shape and time constant confirms that these receptor properties also depend on the interaction between residues 153 and 199.

The rationale behind using neuro 2a cells for muscle nAChR testing is an easy transfection, fast growing nature of these cells [32,33], and a lack of natively expressed acetylcholine receptors [7]. They also are of the same origin with muscle nAChR subunits genes used in the current study—*Mus musculus*—in contrast to human HEK 293T cells, which are routinely used in electrophysiology and calcium imaging. Nevertheless, we could not exclude the possibility that in native muscle fibers nAChRs would show slightly different behavior and this fact should be taken into account when interpreting the results presented in this paper.

Based on results discussed above, we wondered what might be the reason for the G153S mutant to have such a drastic effect on the acetylcholine-evoked calcium ions accumulation, as compared to WT muscle nAChR? To address this question we performed patch-clamp experiments to analyze the shape of ionic charges flow across the membrane of muscle nAChR-expressing cells during EC_100_ acetylcholine (20 μM) application (Figure 5a). The average current amplitude was higher in G153S than in the WT receptor (Figure 5b). Interestingly, the current shapes of G153S mutant and WT muscle nAChR showed differences in decay time constants. To measure this effect quantitatively, we performed a non linear curve fit of averaged current traces for both G153S and WT using a double exponent equation (Figure 5c). Indeed, time constants demonstrated significant differences, G153S-mediated current decaying about three times faster than with WT. Faster current decay lead to lower net charge flow through the ionic channel, which also might contribute to clinical manifestations of SCCMS. However, a faster decay of G153S-mediated acetylcholine-evoked current did not seem to compensate prolongation in Ca^2+^ signal (Figure 4d), which might be caused by a combination of higher potency of acetylcholine toward G153S (Figure 4a, Table 1) and higher average amplitudes of acetylcholine-evoked currents (Figure 5b). Moreover, concentration of acetylcholine in the synaptic cleft decreased faster than current through the receptor decays [33]. Thus higher amplitudes could have greater impact on the physiology than prolongation of the receptor activation in case of G153S slow-channel myasthenia.

## 4. Materials and Methods

### 4.1. Molecular Modeling

The structure of *Torpedo* nAChR (PDB 6UWZ published in [20]) was used as a source of α1 nAChR extracellular domain atomic coordinates. Point mutations were introduced to positions 153 and 199 using UCSF Chimera “Rotamers” tool with the Dunbrack’s rotamer library [34,35]. The Gromos54a7 forcefield [36] was used in GROMACS 5.0.7 [37] to generate the topology and coordinates files. A total SPC water molecules of 18,321-18,324 and seven sodium ions were added to the simulation dodecahedron box to produce periodic boundary conditioned system with 1.2 nm between images. Resulting structures of α1 nAChR extracellular domains were subjected to energy minimization using steepest descend algorithm to reach maximum force < 1000.0 kJ/mol/nm. NPT and NVT 100 ps of heavy atoms-constrained molecular dynamics equilibration steps were followed by 5 ns of unconstrained molecular dynamics. Atomic trajectories were then analyzed with hbdist and hbang instruments provided by the GROMACS package. Results were visualized with the Xmgrace program.

### 4.2. Point Mutagenesis and Transient Transfection

Site-directed mutagenesis was performed using QuikChange kit (Agilent, Santa Clara, CA, USA) as previosly described [7]. Two primer pairs were designed: L219A-F (5′-CCC TAC GCG GAC ATC ACC-3′), L219A-R (5′-GAT GTC CGC GTA GGG AGT G-3′), L219T-F (5′-CCC TAC ACG GAC ATC ACC TAC-3′), L219T-R (5′-GAT GTC CGT GTA GGG AGT GG-3′). Primers were applied to pRBG plasmid containing either WT or G153S murine α1 nAChR. PCR was performed as recommended by manufacturer. Clones obtained as a result of the mutagenesis procedure were sequenced and it was confirmed that target mutations are present.

Murine neuroblastoma neuro 2a cells were obtained from the American Type Tissue Culture Collection (ATCC, Molsheim, France) and maintained as described previously [7]. One day before the transfection, cells were applied to coverslips and placed inside 35-mm cell culture dishes with 2 mL of medium. Transfection medium contained 3 g/1 cDNA of muscle nAChR, kindly provided by Prof. Dr V. Witzemann, and the Lipofectamine 2000 transfection reagent (Invitrogen, Thermo Fisher Scientific, Waltham, MA, USA). Case12-encoding plasmid (0.5 g/l, Evrogen, Russia) was added to transfection medium to facilitate the transfected cells identification. The cDNA-containing solution was replaced by cell culture medium 6-8 h after the transfection. 

### 4.3. Calcium Fluorescent Imaging

To measure changes in intracellular calcium concentration, transfected cells (Neuro2acells) were cultured on polylysine-coated coverslips. Cells were perfused at room temperature with buffer containing 140 mM NaCl, 2 mM CaCl 2, 2.8 mM KCl, 4 mM MgCl 2, 20 mM HEPES buffer, 10 mM glucose, pH 7.4. Expression of Case12, a fluorescent genetically encoded biosensor of calcium ions, allowed direct monitoring of cytoplasmic calcium concentration changes for at least 30 s using a microscope with appropriate filter combination and cooled CCD CAM-XM10 (Olympus, Tokyo, Japan). Case12 has fluorescence in the green region of the spectrum: maxima of excitation and emission of fluorescence at 491 and 516 nm respectively. Coverslips were placed to the recording chamber and washed with continious flow. Calcium responses were evoked by switching to the buffer containing the respective acetylcholine concentration. Videos acquired and processed using CellA Imaging Software (Olympus Soft Imaging Solutions GmbH, Germany) at 200 ms exposure time, Image J and CellX. Cells exposed to acetylcholine (Sigma, Germany) were measured independently using ImageJ Intensity vs. Time monitor.

Cells, cultured in TC-coated polystyrene black 96-well plates (SPL Life Sciences Co. Ltd., Korean Republic) were transfected as described above. Measurements were performed using fluorescence multi-well reader (Hidex, Turku, Finland) as described previosly [7]. Immediately after the acetylcholine application each well was recorded for 80 cycles (2 s each cycle). Peak intensities minus averaged basal fluorescence level were expressed as a percentage of the maximal response obtained.

It is known that some nAChR subtypes benefit from co-expression with chaperones NACHO and Ric-3. it is known that “NACHO mRNA is uniquely enriched in the brain” [37]. Electrophysiological recordings from acutely isolated hippocampal neurons showed complete absence of α7-mediated currents and specific alpha-bungarotoxin binding in brains of NACHO knockouts [38]. Still, no bright muscular abnormalities have been reported in the same organisms, leading us to conclude that NACHO is not an absolute prerequisite for muscle nAChR expression.

Contrary to NACHO, Ric-3 mRNA has been detected in muscle tissue in significant amounts [39]. Ric-3 knock-out mice “showed no overt phenotype” [40], thus confirming that no severe muscular dysfunction, which inevitably would arise as a result of muscle nAChR deficit, has been detected.

In our hands, muscle nAChR expression in either HEK 293-T or neuro 2a did not require an additional boost from chaperon coexpression (see Appendix A). In our experiments we did not use any chaperons.

The possibility of different muscle nAChR subunit combination expression in the model system has been thoroughly discussed in the literature. Incomplete sets of subunits have mostly been shown to lack any functional activity despite some residual alpha-bungarotoxin binding [41,42].

Moreover, we had data that complements these published observations. In short, domestic sheep (*Ovis aries*) muscle was homogenized in liquid nitrogen and ExtractRNA reagent (Evrogen, Moscow, Russia) was applied to obtain total RNA fraction. Complementary DNA strands were synthetized using the MMLV Revertase kit (Evrogen, Moscow, Russia) and specific primers. nAChR subunits were cloned using pcDNA3.1 plasmids (Thermo Fisher Scientific, Waltham, MA, USA) and transfected to neuro 2a or HEK-293T cells in various combinations with murine muscle nAChR. β1, δ, and ε nAChR subunit clones had missense mutations compared to the respective genome sequences, which prevented subunit expression. The *Ovis aries* α1 subunit clone showed a sequence identical to the published genome sequence. Surface nAChR expression was evaluated by 100 nM Alexa 555 α-bungarotoxin staining. Briefly, the fluorescent toxin was applied for 30 min to cells 36-42 h after the transfection. After the initial incubation with the toxin, cells were washed with the buffer described above. Staining was analyzed using CellA Imaging Software (Olympus Soft Imaging Solutions GmbH, Germany) at 500 ms exposure time. To allow direct comparisons equal exposure times and background correction settings were used for each subunit combination. Coexpression tests showed that only a combination that includes α1 nAChR was expressed. The substitution of any other murine muscle nAChR subunit with the respective mutant *Ovis* subunit lead to non-expressing cells (see Appendix A).

### 4.4. Electrophysiology

Patch-clamp recordings were performed 48–72 h after the transfection using whole-cell scheme on EPC-9 amplifier (HEKA Elektronik, Lambrecht, Germany). External perfusion solution contained 140 mM NaCl, 2 mM CaCl2, 2.8 mM KCl, 4 mM MgCl2, 20 mM HEPES and 10 mM glucose,. Solution pH was adjusted to 7.4 (320–330 mOsm). The patch pipette contained 140 mM CsCl, 6 mM CaCl2,2mM MgCl2,2mM MgATP, 0.4 mM NaGTP, 10 mM HEPES/CsOH, 20 mM BAPTA/KOH with pH adjusted to 7.3. Pipettes were pulled from filament-supplied borosilicate glass capillaries (Harvard Apparatus Ltd., Holliston, MA, USA) using Narishige (Tokyo, Japan) pipette puller. Pipettes had resistances of 5–6 MOhm. Acetylcholine was delivered by FastStep perfusion system (Warner Instruments, Hamden, CT, USA).

### 4.5. Data Analisys

Data was analyzed in R programming language version 3.6.3 using standard libraries and visualized using the ggplot2 library. For a dose–response non linear curve fit the following equation was used: % response = 100/{1 + (EC_50_/[acetylcholine])^n^}. Exponential decay data was fitted with equation y = a × exp(b × x) + c × exp(d × x). All results are presented as mean ± standard error of the mean (S.E.M.), unless otherwise stated. Confidence intervals were calculated as {mean − 1.96 × S.E.M.; mean + 1.96 × S.E.M.}.

## 5. Conclusions

In this article by combining computer modeling, molecular dynamics, mutagenesis, electrophysiology, and calcium imaging we analyzed possible role of hydrogen bonds between muscle nAChR α1 subunit residues 153 and 199. Substitution of G153 residue in wild-type muscle nAChR to serine was associated with slow-channel congenital myasthenic condition and was concluded to modify the hydrogen bonding involving the residue 199 located in the C-loop outer surface. Functional assays confirmed that substitutions of these critical points influenced the acetylcholine potency and desensitization kinetics. Our findings shed light on the genesis of slow-channel myasthenic conditions and can help in new nAChR ligand design.

## Figures and Tables

**Figure 1 molecules-26-01278-f001:**
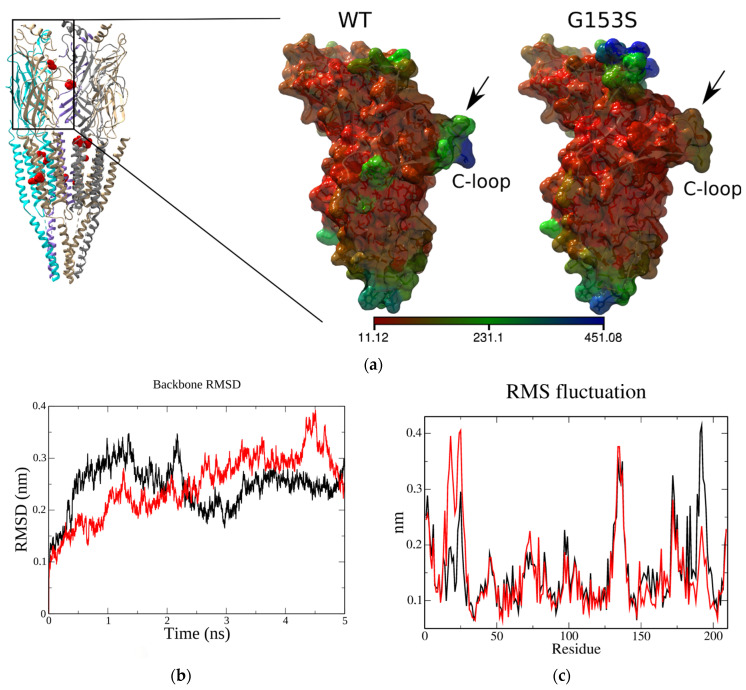
Molecular modeling of the wild type (WT) and G153S mutant *Torpedo* α1 nAChR: (**a**) overall side view of muscle nAChR with amino acid residues, where mutations that are associated with slow-channel myasthenia are shown in red. Note that only one mutation is situated in the extracellular domain of the α1 subunit (the other is situated in the complementary subunit). Atomic coordinates of the extracellular domain of the α1 nAChR subunit have been copied from the PDB 6UWZ and either subjected to molecular dynamics without changing, or after changing of G153 residue to serine. Both molecules underwent 5 ns unconstrained molecular dynamics. The flexibility of amino acid residues side chains of G153S mutant showed moderate differences from the WT. Molecular surfaces are colored according to residue flexibility: the most flexible residues are shown in blue, less flexible are in red (residues of intermediate flexibility are depicted green). Note that the C-loop residues are more rigid in G153S mutant; (**b**) extracellular domains of WT (black) and G153S mutant (red) of α1 nAChR are relatively stable during 5 ns molecular dynamics showing root mean square deviation (RMSD) variability of the backbone in range 0.1–0.2 nm; (**c**) root mean square fluctuations of amino acid residues α-carbons of WT (black) and G153S mutant (red) of α1 nAChR extracellular domains during 5 ns molecular dynamics. N-terminal (10-25) and C-loop (180–200) regions show most prominent difference; (**d**) in the WT receptor the backbone oxygen of G153 is able to form a hydrogen bond with the backbone of L199 residing in the C-loop; (**e**) in the G153S mutant this hydrogen bond switches from the backbone oxygen to the side chain oxygen of serine, which changes the properties of the receptor and lead is to the SCCMS manifestation; (**f**) hydrogen bond angles distribution between the G153 and L199 in the WT receptor (red) compared to the G153S mutant (black); and (**g**) hydrogen bond distances distribution between the G153 and L199 in the WT receptor (red) compared to the G153S mutant (black).

**Figure 2 molecules-26-01278-f002:**
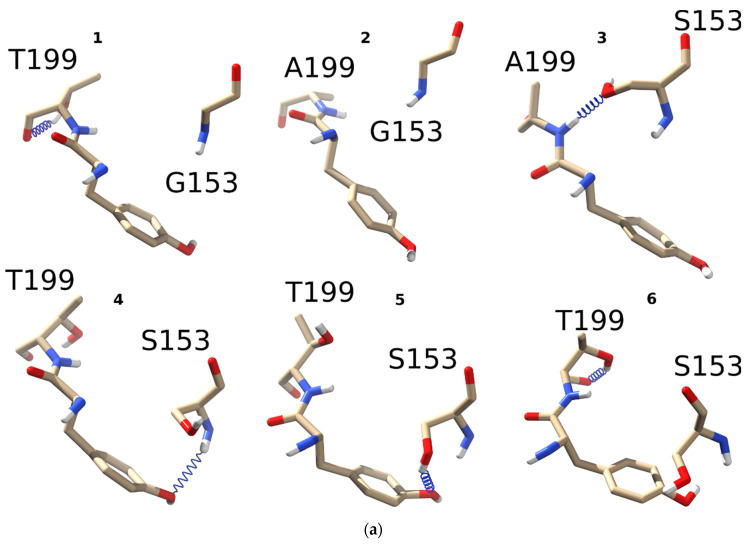
Hydrogen bond patterns formed by threonine, alanine, or leucine in the 199 position of the C-loop with either glycine or serine in position 153: (**a**) four different α1 nAChR mutants were generated in silico, namely L199T (**1**), L199A (**2**), L199A/G153S (**3**), and L199T/G153S (**4**-**6**). All four mutants underwent 5 ns molecular dynamics to monitor hydrogen bond formation between residues 153 and 199. All possible combinations and observed hydrogen bonds (blue spring) are depicted in the figure. Among all mutants L199A did not show any hydrogen bonds between G153 and A199, while L199T/G153S showed three different hydrogen bonds; (**b**) comparison of hydrogen bonds distribution between residues 153 and 199 in double mutants L199A/G153S (black bars) and L199T/G153S (red bars). Distribution of H-bonds by the donor–acceptor angle shows the most prominent difference between these two mutants implying the possibility of different functional properties.

**Figure 3 molecules-26-01278-f003:**
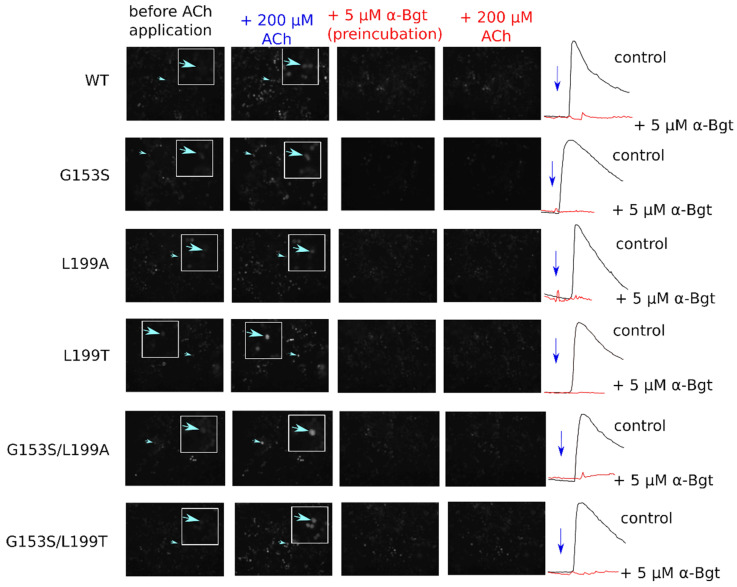
Fluorescent calcium imaging of muscle nAChR-expressing cells. To confirm nAChR expression in transfected cells calcium transients were selectively inhibited by α-bungarotoxin (α-Bgt) preincubation. No transients were recorded from cells treated with α-Bgt. Active cells where representative traces are shown on the figure are pointed by arrows.

**Figure 4 molecules-26-01278-f004:**
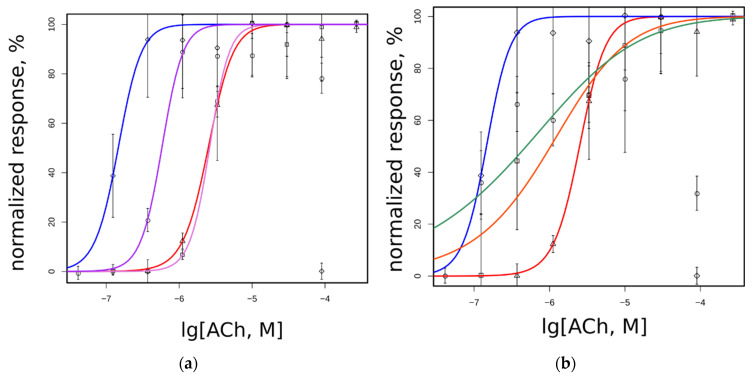
Functional properties of WT and mutant muscle nAChRs studied in the fluorescent cytoplasmic Ca^2+^ assay using genetically encoded sensor Case12: (**a**) dose response curves of L199T (squares, magenta line), L199T/G153S (circles, purple line), WT (triangle, red line), and G153S (diamonds, blue line); (**b**) dose response curves of L199A (squares, orange line), L199A/G153S (circles, green line), WT (triangle, red line), and G153S (diamonds, blue line); and (**c**) fluorescent acetylcholine-evoked response measured using epifluorescent microscope set-up normalized to peak amplitude achieved on each receptor variant. Data given as mean ± 95% confidence interval (*n* = 10). Color code is presented on the figure; (**d**) decay of fluorescent 200 μM acetylcholine-evoked response measured right after the peak. Each point represents the mean of three independent experiments ± standard error.

**Figure 5 molecules-26-01278-f005:**
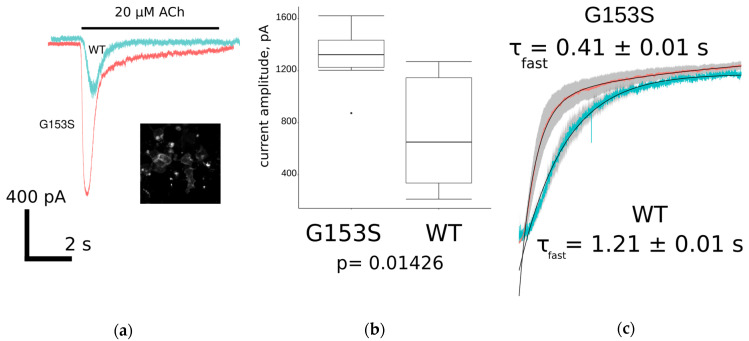
Patch-clamp electrophysiology study of WT and G153S muscle nAChR overexpressed in neuro 2a cells: (**a**) comparison of ionic current trace shapes of WT (light blue) and G153S (pink). Inset shows specific membrane staining with selective muscle nAChR ligand Alexa Fluor 555 α-bungarotoxin; (**b**) comparison of the maximal achieved acetylcholine-evoked current amplitudes for G153S and WT muscle nAChRs represented as a box-and-whisker plot. Mean amplitude of G153S mutant is slightly higher than amplitude of WT receptor at significance level *p* < 0.05 (two-sample *t*-test, *n* = 8); and (**c**) averaged (*n* = 8) current decay of G153S (pink) and WT (light blue) fitted with double exponential curves. Standard errors of the mean values in each time point are shown as gray shadows under the curves. Time constants of fast exponential component differ about three-fold.

**Table 1 molecules-26-01278-t001:** Comparison of muscle nAChR mutants functional properties observed in the fluorescent Ca^2+^ assay.

Name	EC_50_	Time Constant, s
WT	2503 nM (2268 nM, 2763 nM) ^1^	15.0 (14.6, 15.4)
L199A	1035 nM (488 nM, 2199 nM)	19.2 (18.9, 19.5)
L199T	2607 nM (2341 nM, 2903 nM)	14.2 (13.9, 14.5)
L199A/G153S	491 nM (176 nM, 1369 nM)	14.3 (14.0, 14.6)
L199T/G153S	579 nM (392 nM, 856 nM)	11.1 (10.8, 11.4)
G153S	146 nM (122 nM, 174 nM)	34.0 (33.6, 34.4)

^1^ 95% confidence intervals are shown in round brackets.

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
