# Peer review of "Point Mutations of Nicotinic Receptor ?1 Subunit Reveal New Molecular Features of G153S Slow-Channel Myasthenia"

_molecules, 2021, doi:10.3390/molecules26051278_

Round 1

Reviewer 1 Report

This is an interesting study that explores the acetylcholine potency and kinetics of desensitisation of a nAChR mutation G153S, which adds to our understanding of slow channel myasthenic syndromes as well nAChR gating mechanisms. Overall the study is well done and well presented. There are some minor typographical errors / minor corrections that must be addressed.

Page 2 / Line 67-68: The sentence "One of the most crucial regions of α sub-67 units is the loop C which motions upon binding of diverse ligands direct either nAChR 68 activation or inhibition" is not clear and should be paraphrased.

Figure 1: The panel numbering must be corrected, as panel (b) is missing. Accordingly, it may be necessary to amend the figure legend as well.

Page 7 / Line 206: The needs to be consistency in abbreviation. The phrase 'nicotinic acetylcholine receptor" has already previously been abbreviated to nAChR. 

Page 7 / Line 207-208: The work 'uncharacterized' need not be hyphenated as 'un-characterized.' This sentence also needs to end in '....such experiments.' 

Page 8 / Line 265: N=8.....this needs to be 'n=8.'

Page 12 / Line 465: the phrase 'by combining' is repeated twice.

Author Response

We thank the Reviewer 1 for the comments.

Page 2 / Line 67-68 (new file lines 70+): the sentence has been rewritten: "One of the most crucial regions of α subunits is the loop C:  the X-ray structures  of the AChBP complexes with  nicotine (agonist) and such  competitive antagonists as α-cobratoxin,  α-bungarotoxin  and α-conotoxin demonstrated  that an agonist is embraced by this loop moving to the centre of the molecule, while  with antagonists it is shifted to the periphery of molecule."

Figure 1: The panel numbering has been corrected.

Page 7 / Line 206 (new numbering 217) notation has been unified. Abbreviation nAChR is introduced once in each section.

Page 7 / Line 207-208 (new file 212-213): corrected

Page 8 / Line 265 (new file 272): corrected

Page 12 / Line 465 (new file 469): corected

Reviewer 2 Report

The Study by Kudryavtsev and colleagues show a molecular mechanism underlying G135S a1-nAChR mutation-induced abnormal channel functions, which is associated with SCCMS. Authors are particularly interested in the mutation near to the acetylcholine binding site. Using Ca2+ imaging and electrophysiology, authors show that the G153 mutation modifies hydrogen bonds in the receptor, thus influences a1-nAChR functions. The experiments are designed well, and the conclusion is fine. However, there are some points need to be clarified.

  1. It has been shown that many subtypes of nAChRs need chaperon proteins, such as NACHO or Ric-3, to be fully functional in non-neuronal cells. Does alpha1 need chaperon proteins to be functional in neuroblastoma cells? If it does, authors need to repeat the experiments with co-expression of chaperon proteins.
  2. In general, muscle nAChRs include alpha1, beta1, delta, and gamma subunits. Therefore, expression of only alpha1 subunit in cells generate alpha1 homopentamer form of nAChRs, which do not represent the physiological conformation. Authors need to discuss the potential problems by using homopentameric receptors for their experiments.
  3. In Fig. 3 and 4, authors need to show the receptor expression and localization in cells.
  4. In Fig. 3, authors need to show representative tracer in each condition. Also, the method is not clear. How long did author record the signals? What is the exposure time? What is the interval between the images?
  5. In Fig. 3 and 4, how did author deliver acetylcholine?
  6. In Fig.4, the amplitude of WT and G153S is the same, but the graph shows significant increase. Authors need to show better representative traces.
  7. Why did author choose N2a cells to express muscle nAChRs? Would it be possible that the mutant acts differently in muscle cells?

Author Response

We thank the Reviewer 2 for the questions and remarks.

1) It has been shown that many subtypes of nAChRs need chaperon proteins, such as NACHO or Ric-3, to be fully functional in non-neuronal cells. Does alpha1 need chaperon proteins to be functional in neuroblastoma cells? If it does, authors need to repeat the experiments with co-expression of chaperon proteins.

Answer:

Whether muscle nAChR requires chaperones for the pentameric assembling is an interesting question. NACHO and Ric-3 info on muscle nAChR possible association is not fully disclosed in literature yet. However, it is known that "NACHO mRNA is uniquely enriched in the brain" (https://www.sciencedirect.com/science/article/pii/S0896627316000192). Electrophysiological recordings from acutely isolated hippocampal neurons showed complete absence of α7-mediated currents and specific alpha-bungarotoxin binding in brains of NACHO knockouts (https://www.sciencedirect.com/science/article/pii/S0896627316000192). Still, no bright muscular abnormalities have been reported in the same organisms, leading us to conclude that NACHO is not an absolute prerequisite for muscle nAChR expression.
Contrary to NACHO, Ric-3 mRNA has been detected in muscle tissue in significant amounts (https://www.sciencedirect.com/science/article/pii/S0021925820837582?via%3Dihub). Ric-3 knock-out mice "showed no overt phenotype" (https://www.ncbi.nlm.nih.gov/pmc/articles/PMC7175337/), thus confirming that no severe muscular dysfunction, which inevitably would arise as a result of muscle nAChR deficit, has been detected.
In our hands, muscle nAChR expression in either HEK 293-T or neuro 2a does not require an additional boost from chaperone co-expression (see Fig. S1).
We recognise that muscle nAChR and chaperones interaction is indeed important topic. However, the addition of one or two plasmids with the chaperone genes on top of four plasmids encoding muscle nAChR genes would deliver excessive complexity to the system without a noticeable gain in molecular mechanisms understanding.
Necessary amendments have been done to the article text to cover this topic and inform our readers about the current study's limitations.

2) In general, muscle nAChRs include alpha1, beta1, delta, and gamma subunits. Therefore, expression of only alpha1 subunit in cells generates an alpha1 homopentamer form of nAChRs, which do not represent the physiological conformation. Authors need to discuss the potential problems by using homopentameric receptors for their experiments.

Answer:

The possibility of different muscle nAChR subunit combination expression in the model system has been thoroughly discussed in the literature. Incomplete sets of subunits have mostly been shown to lack any functional activity despite some residual alpha-bungarotoxin binding (https://febs.onlinelibrary.wiley.com/doi/epdf/10.1016/0014-5793%2887%2980065-0).
Moreover, we have data that complements these published observations. In short, domestic sheep (Ovis aries) muscle nAChR subunits has been cloned and transfected to neuro 2a or HEK-293T cells in various combinations with murine muscle nAChR. Beta1, delta and epsilon nAChR subunit clones had missense mutations compared to the respective genome sequences which prevented subunit expression. Ovis aries alpha1 subunit clone showed sequence identical to the published genome sequence. Co-expression tests showed that only a combination that includes alpha1 nAChR is expressed. The substitution of any other murine muscle nAChR subunit with the respective mutant Ovis subunit leads to non-expressing cells (see S2).

3) In Fig. 3 and 4, authors need to show the receptor expression and localization in cells.

A new Figure 3 has been added to show control experiments confirming muscle nAChR expression. Figure 4 (Figure 5 in the revised manuscript) has been amended with the inset picture to show receptor expression and localisation.

4) In Fig. 3, authors need to show representative tracer in each condition. Also, the method is not clear. How long did the author record the signals? What is the exposure time? What is the interval between the images?

Traces have been added to the figure 3. "Methods" section has been edited (lines 443-470).

5) In Fig. 3 and 4, how did author deliver acetylcholine?

"Methods" section has been edited: lines 443-470 (calcium imaging) and line 467 (electrophysiology).

6) In Fig.4, the amplitude of WT and G153S is the same, but the graph shows significant increase. Authors need to show better representative traces.

Traces have been replaced.

7) Why did author choose N2a cells to express muscle nAChRs? Would it be possible that the mutant acts differently in muscle cells?

Answer:

The rationale behind using neuro 2a cells for muscle nAChR testing is an easy transfection, fast-growing nature of these cells, and a lack of natively expressed acetylcholine receptors. They are also of the same origin - Mus musculus - in contrast to human HEK 293T cells, routinely used in electrophysiology and calcium imaging. Nevertheless, we can not exclude the possibility that in native muscle fibres nAChRs would show slightly different behaviour. The "Discussion" section has been amended to reflect this study limitation (lines 383-388).

Round 2

Reviewer 2 Report

In the revised version of the manuscript the authors have addressed my major concerns, however, there are still some concerns.

  1. The resolution of Fig. 3 should be improved. Also, it would be appreciated to have higher magnification of cell images as an inset.
  2. In the discussion, authors stated “Rationale behind using neuro 2a cells for muscle nAChR testing is an easy transfection, fast growing nature of these cells and lacking of natively expressed acetylcholine”. This needs proper references.
  3. There is no method for Suppl. Figures. ex) The sources of all plasmids, detailed procedure of staining, microscopy, and etc.

Author Response

We thank the reviewers for such constructive commentaries. Here are our answers to the second round of the review:

"The resolution of Fig. 3 should be improved. Also, it would be appreciated to have higher magnification of cell images as an inset."

Answer: Figure resolution is 4813 to 3815 pixels, and its apparent poor quality is probably due to PDF image compression issues. The publisher would fix it. Higher magnification insets are now provided in Fig. 3 as requested.

"In the discussion, authors stated “Rationale behind using neuro 2a cells for muscle nAChR testing is an easy transfection, fast growing nature of these cells and lacking natively expressed acetylcholine”. This needs proper references."

Answer: References have been added (see lines 388-389). It should be noted that this test system's extensive characterisation based on N2A cells has been disclosed by Shelukhina et al. in ref [7]. We are pretty sure that no significant influences on nAChR functioning in transfected cells take place.

"There is no method for Suppl. Figures. ex) The sources of all plasmids, detailed procedure of staining, microscopy, and etc."

Answer: Methods section have been amended. See lines 490-505.